# Aqueous Extracts of Lemon Basil Straw as Chemical Stimulator for Gray Oyster Mushroom Cultivation

**DOI:** 10.3390/foods11091370

**Published:** 2022-05-09

**Authors:** Pragatsawat Chanprapai, Thanaporn Wichai, Sarintip Sooksai, Sajee Noitang, Weradaj Sukaead, Winatta Sakdasri, Ruengwit Sawangkeaw

**Affiliations:** 1The Institute of Biotechnology and Genetic Engineering, Chulalongkorn University, 254 Phayathai Road, Pathumwan, Bangkok 10330, Thailand; pragatsawat.s@chula.ac.th (P.C.); numaoy.w@chula.ac.th (T.W.); sarintip.so@chula.ac.th (S.S.); sajee.no@chula.ac.th (S.N.); weradej.s@chula.ac.th (W.S.); 2Program in Food Process Engineering, School of Food-Industry, King Mongkut’s Institute of Technology Ladkrabang, 1 Chalong Krung 1 Alley, Lat Krabang, Bangkok 10520, Thailand; winatta.sa@kmitl.ac.th; 3Research Unit in Bioconversion/Bioseparation for Value-Added Chemical Production, Chulalongkorn University, 254 Phayathai Road, Pathumwan, Bangkok 10330, Thailand

**Keywords:** biological efficiency, fruiting body, gray oyster mushroom, lemon basil, mushroom cultivation, *Ocimum*, *Pleurotus*

## Abstract

To reduce the burning of lemon basil straw (LBS)—the byproduct of basil seed production—we propose utilizing LBS as a replacement substrate for mushroom cultivation. LBS can stimulate both mycelial growth and percentage biological efficiency; however, the rigidity of this material limits particle size reduction. In this work, aqueous extractions were facilely performed without using either hazardous chemicals or complex procedures to valorize LBS as a stimulator for gray oyster mushroom cultivation. An aqueous extraction at solid-to-liquid of 50 g/L was employed. The macerated-LBS and decocted-LBS extracts were tested for mycelial growth in potato dextrose agar and sorghum grains. Following this, both aqueous extracts were applied as a wetting agent in cylindrical baglog cultivation to estimate mycelial growth, biological efficiency, and productivity. It was found that LBS extracts insignificantly enhanced the mycelia growth rate on all media, while the diluted LBS (1:1 *v*/*v*) extracts improved 1.5-fold of percentage biological efficiency. Gas chromatograph-mass spectrometer results indicated 9-octadecaenamide is a major component in LBS aqueous extract. Results demonstrated that the LBS extract is a good stimulator for the production of *Pleurotus* mushroom.

## 1. Introduction

Lemon basil (*Ocimum citriodorum* Vis., Synonym. *Ocimum* × *africanum*) seed—hereafter referred to as “basil seed”—is often used to make a traditional dessert in Asian countries. For the dessert, water-soaked basil seeds are mixed with jelly and either fruit juice or milk. Basil seed mucilage—also called basil seed gum—provides numerous health benefits, including blood sugar control and cholesterol level reduction [1]. Every kilogram of harvested basil seed generates 1–1.5 kg of basil straw, which is a waste product that is often left to rot or burned in open fields. In Thailand, basil seed production generated over 350 tons of lemon basil straw (LBS) in 2018 [2].

Mushroom cultivation requires a growth substrate, and the development of alternative substrates has been a subject of research worldwide. Mostly, those substrates are agricultural wastes, such as rice and wheat straw [3,4], and food wastes, such as waste tea leaves [5], spent coffee grounds [6], and olive mill residue [7]. LBS has also been investigated as a replacement substrate, and a previous study [8] found that for gray oyster mushroom (*Pleurotus Sajor-Caju* (Fr.) Sing.) cultivation, the optimum mass ratio of LBS is 50 wt% (dry basis). At this percentage, LBS can reduce the cost of growing medium (“baglog”) production by 25%. The LBS also promoted the mycelial growth, biological efficiency, and antioxidant activity of the fruit body. However, their results showed that LBS is difficult to manage when compared to sawdust. For example, improperly milled LBS could potentially pierce through the plastic baglog because of its hardness and sharpness. Additionally, particle size reduction of LBS—i.e., chopping, milling, and sieving—are energy, time, and labor intensive. Furthermore, the low bulk density of LBS posed potential negative effects on mycelial growth rate. 

For the other studies, various physical and chemical treatments were demonstrated to promote mycelial growth and an increased production yield of edible mushrooms [9,10,11,12,13]. However, the demonstrations were confronted by numerous individual barriers. The physical treatments for the study require specific equipment to generate pulsed electric fields [9] or ultrasonic and acoustic sound [10]. This infers that the economic feasibility increases together with both the production capacity and/or scale of farming. Furthermore, the addition of a pure compound is practical for small-scale farming. For example, an optimal concentration of ascorbic acid accelerated the mycelial growth of *P. ostreatus* by promoting ascorbate oxidase activity [11]. The multi-component stimulators also required additional post-treatments to maximize stimulating activity. For instance, crude wood vinegars reduced the mycelial growth of *P. ostreatus*, while distilled wood vinegars (boiling, point range of 100 °C to 105 °C) improved its growth rate by up to 185% that of the control medium [12]. The macerated ethanolic extract of *Agave potatorum* Zucc. leaves needed to be concentrated in a vacuum evaporator before being used as a stimulator at the optimal concentration of 1000 ppm [13].

Aqueous extraction, such as leaching, macerating, and decocting, is a simple method for extracting water-soluble compounds in biomass to use as a growth stimulator of edible mushrooms. For example, 20 g/L of decocted vermicompost, 25 day fermented wheat straw, and dry sheep manure could promote the mycelial growth of *Pleurotus* spp. A vermiwash produced by water leaching and macerating vermicompost for 2 weeks also stimulated the mycelial growth [14]. Moreover, the 100 g/L of decocted corn stover increased the mycelial growth rate and nutritional value in fruit bodies of both *P. ostreatus* and *P. pulmonarus*, while the aqueous extracts of barley and wheat straws did not increase either [15].

The main aim of this research was to simplify the process of utilizing LBS as a growth stimulator for gray oyster mushroom cultivation through aqueous maceration and decoction at atmospheric pressure. The use of harmful chemicals, auxiliary equipment, and complex procedures could be avoided to enable ordinary farmers to adopt the process. Aqueous LBS extracts can be used locally by lemon basil farmers or distributed to mushroom farmers. Analysis of the aqueous extract by gas chromatography–mass spectrometry (GC–MS) identified the active compounds responsible for influencing mycelial growth, biological efficiency, and productivity. Among these compounds is caryophyllene oxide [16], an alternative substrate for mushroom cultivation [8] and gray oyster mushroom mycelial production [17]. Valorization of LBS will discourage farmers from burning this material. In this study, we have utilized LBS aqueous extract as a stimulator for mushroom fruiting. 

## 2. Materials and Methods

### 2.1. Materials

The LBS sample was collected from a basil farm in Si Samrong district, Sukhothai Province, Thailand, in January 2022. Details of the LBS sampling method were adopted as per previous studies, which included specifications for growing condition and sampling sites [8,16]. The LBS sample was used in the condition as it was received—i.e., straight from the sample—without milling or any further treatment conducted. Similarly, mushroom samples were collected, identified, and isolated as mentioned in previous studies [8,17]. All common chemicals required for preparing culture media, e.g., dextrose, agar, and yeast extracts, were purchased from SAC SCI-ENG Limited Partnership, Thailand. Potato and sorghum grain were procured from a local farmer. The para rubber (*Hevea brasiliensis*) sawdust and supplementary materials—e.g., rice (*Oryza sativa*) bran, MgSO_4_, and CaCO_3_—used for the production of cylindrical baglog were obtained from local suppliers for mushroom agriculturists.

### 2.2. Aqueous Extractions of Lemon Basil Straw

The 500 g of collected and dried LBS (10.5 wt% of moisture) was macerated in 10 L of deionized water at room temperature (30 °C ± 2 °C) for 72 h. A similar solid-to-liquid mass ratio was decocted at 100 °C for 15 min. Following this, the hot-decocted water was quenched in an ice-water bath until room temperature, over 15 min. Both extracts were filtrated using a filter cloth followed by filter paper (Whatman No. 1) to remove fine particles. They were then kept in a 20 L polypropylene gallon at 4 °C before using them in further experiments. Thereafter, LBS extracts macerated at room temperature and decocted at 100 °C were referred to as “M-LBT” and “D-LBS,” respectively. The abbreviations M-LBS (1:1) is M-LBT mixed with water (1:1, *v/v*), and D-LBS (1:1) is D-LBS mixed water (1:1, *v/v*). 

### 2.3. Mycelial Cultivation on Potato Dextros Agar and Sorghum Grain

To determine the effects of LBS extracts on mycelium growth, mycelial cultivation was tested in a petri dish. Potato dextrose agar (PDA) was prepared from 20.00 g of dextrose, 15.00 g of agar, and 200 g of potato and infused in 1 L of deionized water (DI). The M-LBT and D-LBS extracts were added at a volumetric ratio of 1:1 as treatments when compared with pure PDA. Sub-cultured mycelia (5 mm diameter) were cocked at the center of the petri dishes in six replicates. All samples were incubated at room temperature (30 °C ± 2 °C) for a period of 10 days. The diameter of the colony was measured daily to estimate the growth rate.

The effects of LBS extracts on mycelium growth were also investigated in a sorghum grain (SG) medium. The SG medium was washed with DI water before being soaked for 12 h. The SG and DI water mixture (volumetric ratio of 1:1) was boiled for 20 min and used as a positive control [18]. The M-LBT, M-LBT (1:1), D-LBS, and D-LBS (1:1) extracts were replaced with DI water in a boiling SG medium to conduct as four individual treatments. A 35-g amount of all media was spread on a petri dish and sterilized. After cooling, 5 mm diameter of sub-cultured mycelia was inoculated in 6 replicates. All treatments were incubated at room temperature (30 °C ± 2 °C) in darkness for a total of 8 days. The diameter of mycelium growth was recorded every two days. The mycelial growth rates on both PDA and SG media were averaged from three subculture generations.

### 2.4. Cylindrical Baglog Production, Fruiting, and Harvesting

Rubber sawdust (<1.0 mm.) was mixed with 9.00% rice bran, 0.03% MgSO_4_, and 0.03% CaCO_3_, calculated by weight of sawdust and used as a core substrate. Total nitrogen and carbon contents of rubber sawdust were 0.50 wt% and 33.90 wt%, respectively. The moisture contents in substrates were adjusted to 60–65 wt% by adding only tap water as a negative control. The four treatments conducted were: a) M-LBT, b) M-LBT (1:1), c) D-LBS, and d) D-LBS (1:1) used to regulate the moisture content instead of tap water. The total volume of added water or LBS extracts was ~4 L per 10.0 kg of the moist substrate. All experiments were conducted in duplicate by using 20 baglogs (~0.5 kg, 6.5 cm ID × 6.5 cm length) for each treatment. The sterilized substrate baglogs were inoculated using seed spawns of *P. sajor-caju* (Fr.) Singer, tightened with a rubber band and closed using a plastic vent cap. All baglogs were incubated at room temperature (24–30 °C and 60–70% RH) in a shaded area for 15 days. The mycelial growth was examined using average-length three-strip markers on substrate bags every two to three days. Fruiting and harvesting conditions were identical to those described in previous work [8]. The percentage biological efficiency (%BE) and productivity were calculated from the total weight of harvested fresh mushrooms over 30 days.

### 2.5. Gas Chromatograph–Mass Spectrometer Analysis

The gas chromatograph (GS; Agilent, GC6890N, Bangkok, Thailand) and the mass selective detector (Agilent, 5975 MSD) equipped with a capillary column (HP-5 ms, 0.25 mm ID × 0.25 µm × 30.0 m) were employed to identify the bioactive compounds present in aqueous extract. Ultra-pure helium (99.999%) was used as a carrier gas at a constant flow rate of 1.0 mL/min. The split–splitless injector (Agilent, 7633 ALS) was set at a split ratio of 1:5, with an injection volume of 1.8 μL, and set at a constant temperature of 250 °C. The column oven was programmed at an initial temperature of 50 °C, initial time of 2 min, final temperature of 280 °C, a temperature ramps rate of 5 °C/min, and a final time of 5 min. After 53 min of analysis, the column temperature and flow rate were ramped up to 300 °C and 2 mL/min, respectively, for post-treatment. 

The mass spectrometer (MS) transfer line and ion source temperatures were set at 280 °C and 230 °C, respectively, whereas the ion source voltage was set to 70 eV. Solvent delay was set at 2 min. A molecular weight scanning range of 33–500 m/z was employed. The compounds were identified by comparison of sample mass spectrum using the NIST2011 library. The aqueous extract (0.5 wt%) was diluted in absolute ethanol (99.995%) to a concentration of 500 ppm before analysis was conducted in order to reduce the interfering effect of moisture on the filament of the MS ion source. Compound amounts were quantified by using the total sum normalization method [19].

### 2.6. Statistical Analysis

Results from this study are presented in terms of means ± standard deviations from three different experiments. All data were analyzed using Tukey pairwise comparisons, with significant differences accepted at the *p* < 0.05 level on the Minitab^®^ v20.3 for Windows, trial version, Minitab, LLC, State College, PA, USA.

## 3. Results

### 3.1. Effects of Lemon Basil Straw Extracts on Mycelial Growth

Table 1 shows the mycelial growth rates of *P. sajor-caju* in differentially solid media kept at room temperature. 

The appearances of mycelia grown on PDA and SG media are shown in Figure 1. The maximum mycelial growth rate in the range of 12.15–13.33 mm/day was observed for the SG medium. Alternately, the cultivation on the cylindrical baglog (CB) media revealed the lowest mycelial growth rate as well as the highest standard deviation. The mycelium in CB was fully colonized within 15 days. The appearance of mycelia growing on CB medium is shown in Figure 2.

### 3.2. Effects of Lemon Basil Straw Extracts on Biological Efficiency

Table 2 presents the weight of a fresh mushroom (W_FM_), weight of a dry substrate (W_DS_), %BE, and productivity of the cultivated mushroom. Results in the table show that maximum %BE and productivity are obtained by using M-LBS (1:1) and D-LBS (1:1) as a growth simulator. On the other hand, pure M-LBS and D-LBS significantly reduced both %BE and productivity in the fruiting state. The SI unit of productivity in ecology is mass per unit volume per day. However, studies on mushroom production were reported in terms of harvested fresh mushroom weight divided by dried substrate weight and number of harvesting days [8,20]. Productivity after 20 days is shown in Figure 3. The maximum productivities of all treatments were found within 15 days after opening the plastic cap.

### 3.3. Gas Chromatograph–Mass Spectrometer (GC–MS) Analysis

Figure 4 illustrates the chromatogram of M-LBS and D-LBS with the identified compounds (%similarity index > 85%). The bioactive compounds found in both extracts at retention times of 22.881–22.878 min and 39.606–39.625 min are 2,6-bis (1,1-dimethylethyl)-phenol and 9-octadecaenamide, respectively.

## 4. Discussion

In this study, the stimulating activity of LBS aqueous extracts acting as a chemical stimulator for gray oyster mushroom cultivation was indicated by mycelial growth rate on three solid media and %BE. The LBS aqueous extracts were obtained through simple methods via maceration (M-LBS) and decoction (D-LBS). To minimize regular changing procedures for mushroom cultivation, aqueous extracts were designed to mix with core substrates as a wetting agent in CB production. After mushroom mycelium was fully colonized for 15 days, %BE was determined accordingly.

In a previous study, an excess amount of milled LBS, over 50 wt% of CB, inhibited mycelial growth in response to low bulk density of LBS increasing the void within CB medium. Chemical compounds—such as linalool, geranial, and caryophyllene oxide—detected in the milled LBS also exhibited antifungal activity [8]. According to Table 1, the mycelial growth rates of control and treatments within PDA, SG, and CB media are not significantly different. Therefore, it is inferred that M-LBS and D-LBS extracts do not inhibit the mycelial growth of *P. sajor-caju*.

According to Figure 4, a chemical compound, 9-octadecenamide, was detected through GC–MS in both M-LBS and D-LBS. However, the antifungal compounds found in milled LBS were not detected in the aqueous extracts, as they are non-polar compounds. The concentration of 9-octadecenamide in the M-LBS extract (17.34 wt%) is lower than the D-LBS extract (59.99 wt%). It has been reported that 9-octadecenamide is an amide derivative of octadecenoic acid or oleic acid (C18:2) and consequently has antimicrobial activity [21]. Even so, the results of this study revealed that 9-octadecenamide does not affect the mycelial growth of *P. sajor-caju*.

As suggested in previous studies, the mycelial growth rate of *Pleurotus* spp. was influenced by the C:N ratio [20]. Nitrogen-rich substrates induce mycelial growth, although excess nitrogen will also inhibit mycelial growth. Such growth can be promoted through the addition of ascorbic acid [11], wood vinegar [12], and ethanolic extracts of *Agave potatorum* Zucc. leaves [13] into the cultivation media. The chemical components influencing the mycelial growth of *P. ostreatus* have been classified into two groups: enhancing compounds (alcohol, ester, and aldehyde) and inhibitory compounds (phenol, ketone, and carboxylic acid) [12]. Among the inhibitory compounds, the lignin-derived phenolic monomer has the strongest effect on the mycelial growth of both *P. ostreatus* [13] and *P. sajor-caju* [22]. As presented in Figure 4, concentrations of 2,6-bis (1,1-dimethylethyl)-Phenol in M-LBS and D-LBS are only 1.36 wt% and 3.14 wt%, respectively. Based on this observation, it could be concluded that maceration and decoction prevent leaching phenolic compounds from LBS.

According to Table 2, it is clear that M-LBS (1:1) and D-LBS (1:1)—the 50% dilution of aqueous extracts—significantly enhanced %BE of *P. sajor-caju*. by 40%, whereas, pure aqueous extracts, M-LBS and D-LBS, reduced %BE by 20%. The %BE of *Pleurotus* spp. cultivated on various plant-derived food residues were reported in a wide range of cultures, namely, of 25–137% [23]. The %BE strongly depends on substrate type, specifically, its nitrogen content, or C:N ratio. In brief, substrates with high nitrogen content, i.e., low C:N ratio, promote %BE. However, an excess in nitrogen could reduce the %BE because of high substrate pH and high laccase activity, meaning, a C:N ratio lower than 30 simultaneously increases laccase production and further reduces %BE [24,25]. For instance, the %BE of *P. ostreatus* was dropped from 50.7% to 42.6% and 14.7% when using 50% (*v/v*) and 100% (*v/v*) of olive oil effluent, with a total nitrogen of 257.5 mg/L as an alternative wetting agent, respectively [26].

The effects of other stimulators on mushroom yield and %BE have been reported. Among physical stimulators, pulsed electric fields (total electrical energy of 0.017 kWh for 30 days) are known to improve the total yield of *P. sajor-caju* by 34.0% as compared to the control group [9]. Under optimal conditions, acoustic and ultrasonic sound treatments on *P. sajor-caju.* can generate %BE values of 76.5% and 61.7%, respectively, whereas a control sample under optimal conditions has a %BE of 58.2% [10]. Among chemical stimulators, the use of maize-processing effluent at a rate of 50% (*v/v*) as an alternative wetting agent for cultivating *P. ostreatus* can increase %BE to 91.12% compared to 41.51% BE when using DI water [27]. Therefore, the stimulatory activity of LBS extract is comparable to that of other stimulators.

The study on mushroom stimulation lacks information on %BE and productivity because the processes of fruiting and harvesting mushrooms are time-, area-, and labor-intensive. Most studies have investigated mycelial growth on PDA or grain media [11,12,13,14,15]. However, the growth of mycelia on PDA and SG are significantly different from that on CB medium, as shown in Table 1. It should be noted that similar mycelial growth rates in production media can result in dissimilar %BE and productivity values (See Table 2). 

## 5. Conclusions

Aqueous maceration and decoction are suitable methods for extracting biological compounds in LBS for use as a growth stimulator in gray oyster mushroom cultivation. LBS extracts did not significantly influence mycelial growth rates; however, they impacted %BE and productivity. Using diluted LBS extracts, 1:1 (*v/v*) as a wetting agent significantly enhanced the %BE. The GC–MS detected 9-octadecenamide in LBS extracts, while it did not find trace amounts of antifungal compounds in LBS extracts. 9-octadecenamide is classified as both a nitrogen source and bioactive compound. The results of this work could be extended through various means; for example, for the aqueous extraction of other agri-food wastes for cultivation of *Pleurotus* spp. and using LBS aqueous extract as a growth stimulator for other edible mushrooms. The solid residue of LBS extracts could then be used as feedstock for the production of biochar.

## Figures and Tables

**Figure 1 foods-11-01370-f001:**
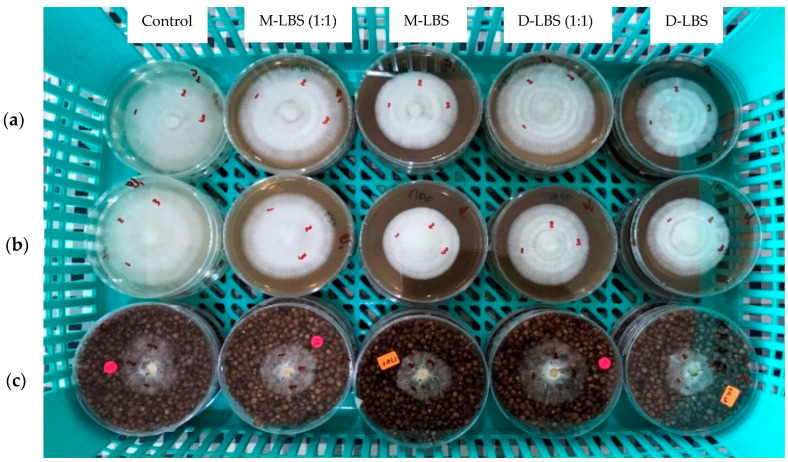
*P. sajor-caju* mycelia growing on (**a**) Potato dextrose agar at 4 days, (**b**) Potato dextrose agar at 3 days, and (**c**) Sorghum grain at 2 days.

**Figure 2 foods-11-01370-f002:**
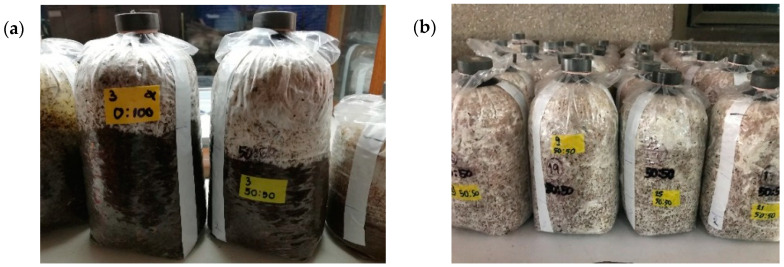
*P. sajor-caju* mycelia growing on cylindrical baglog (**a**) left, M-LBS and right, M-LBT (1:1) at 7 days and (**b**) showing full colonization of four treatments, M-LBS, M-LBS (1:1), D-LBS, and D-LBS (1:1) from left to right.

**Figure 3 foods-11-01370-f003:**
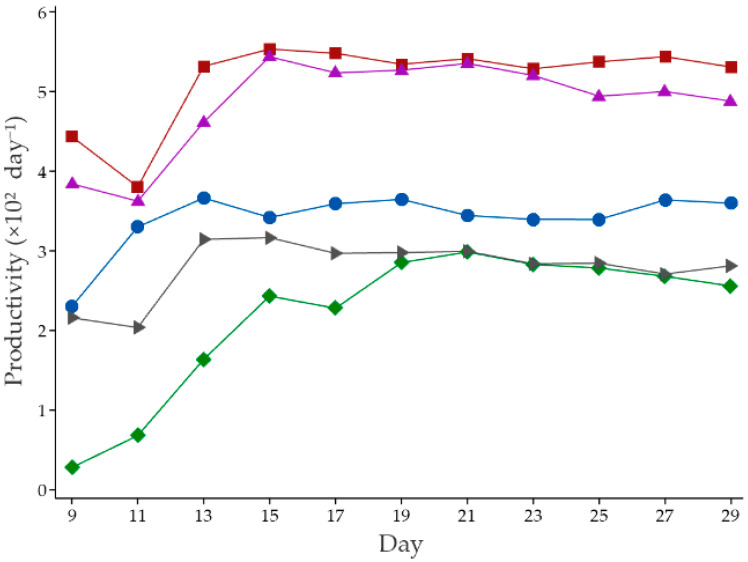
The productivity of gray oyster mushroom cultivated on cylindrical baglog (CB) with various treatments: (●) Control, (■) M-LBS (1:1), (♦) M-LBS, (▲) D-LBS (1:1), and (▶) D-LBS within 30 days.

**Figure 4 foods-11-01370-f004:**
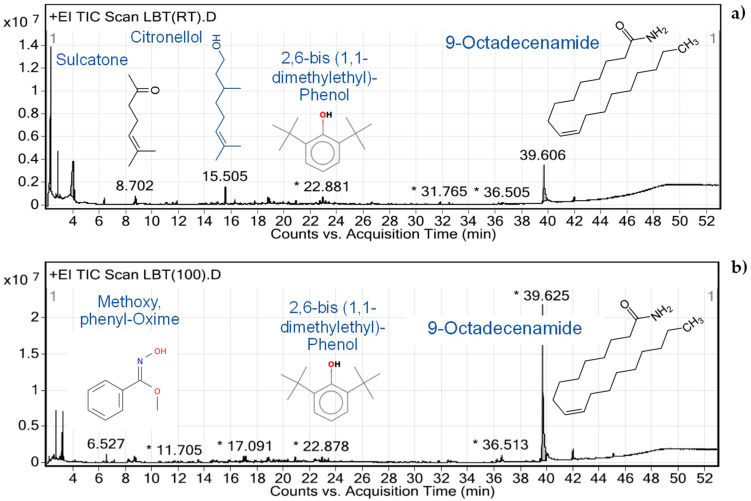
The gas chromatograph–mass spectrometer (GC–MS) chromatograms of: (**a**) M-LBS and (**b**) D-LBS aqueous extracts. * is peak overlapping.

**Table 1 foods-11-01370-t001:** Mycelial growth rate on different solid cultural media at room temperature (30 °C ± 2 °C) for Potato dextrose agar (PDA) and Sorghum grain (SG) and at 24 °C–30 °C for cylindrical baglog (CB).

Treatment	Mycelial Growth Rate (mm/day) ^1^		
	Potato Dextrose Agar (PDA)	Sorghum Grain (SG)	Cylindrical Baglog (CB)
Control	10.12 ± 0.61 ^a^	12.56 ± 1.51 ^a^	7.06 ± 2.88 ^a^
M-LBS (1:1)	10.50 ± 0.46 ^a^	12.40 ± 0.86 ^a^	6.60 ± 2.82 ^a^
M-LBS	9.82 ± 0.67 ^a^	12.60 ± 1.32 ^a^	6.33 ± 2.20 ^a^
D-LBS (1:1)	10.29 ± 0.38 ^a^	13.33 ± 0.49 ^a^	6.04 ± 3.15 ^a^
D-LBS	10.00 ± 0.48 ^a^	12.15 ± 1.19 ^a^	7.73 ± 2.60 ^a^

^1^ Mean values with different superscript letters in each column are significantly different (*p* < 0.05) M-LBS is LBS macerated at room temperature, M-LBS (1:1) is M-LBT mixed with water (1:1, *v/v*), D-LBS is LBS decocted at 100 °C, and D-LBS (1:1) is D-LBS mixed with water (1:1, *v/v*).

**Table 2 foods-11-01370-t002:** The biological efficiency (%BE) and productivity of gray oyster mushroom cultivated on various treatments.

Treatment	W_FM_ (g)	W_DS_ (g)	%BE ^1^	Productivity ^2^ (×10^2^ day^−1^)
Control	1002.30 ± 96.50 ^b^	1392.46 ± 15.96 ^a^	71.98 ± 6.93 ^b^	3.60 ± 0.35 ^b^
M-LBS (1:1)	1497.80 ± 100.00 ^a^	1365.69 ± 13.65 ^a^	109.67 ± 7.33 ^a^	5.48 ± 0.37 ^a^
M-LBS	832.30 ± 18.00 ^c^	1499.61 ± 21.02 ^a^	55.50 ± 1.20 ^c^	2.77 ± 0.06 ^c^
D-LBS (1:1)	1446.80 ± 61.30 ^a^	1412.00 ± 14.04 ^a^	102.46 ± 4.34 ^a^	5.12 ± 0.22 ^a^
D-LBS	806.00 ± 46.40 ^c^	1404.60 ± 11.08 ^a^	57.38 ± 3.31 ^c^	2.87 ± 0.17 ^c^

^1^ Mean values with different superscript letters in each column are significantly different (*p* < 0.05); W_FM_ is a weight of fresh mushroom; W_DS_ is a weight of dry substrate; %BE = (W_FM_/W_DS_) × 100; ^2^ Productivity = W_FM_/(W_DS_ × 20, day of cultivation). M-LBS is LBS macerated at room temperature, M-LBS is LBS macerated at room temperature, M-LBS (1:1) is M-LBT mixed with water (1:1, *v/v*), D-LBS is LBS decocted at 100 °C, and D-LBS (1:1) is D-LBS mixed with water (1:1, *v/v*).

## Data Availability

Data is contained within the article.

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
