# Peer review of "Aqueous Extracts of Lemon Basil Straw as Chemical Stimulator for Gray Oyster Mushroom Cultivation"

_foods, 2022, doi:10.3390/foods11091370_

Round 1

Reviewer 1 Report

ABSTRACT

Please change the order of the first two lines so it can have a better impact

What do the authors mean by “fabricating” in line 21…..seems disconnected from the last idea

INTRODUCTION

Line 43-46 is good but is not in the proper place. Please re-order the referred idea to the final section of the introduction.

Line 47, please add the surname of the author of reference 4 according to MDPI guidance

Line 51, “however, their results”

METHODS

The methods are clear and easy to follow. However, a double-check is always preferred

RESULTS

The addition of a picture of the experiment may improve how the readers can understand the mushroom production

DISCUSSION

The authors must stress the lack of information in this specific area, maybe with a short table or by adding a short paragraph that summarizes the number of scientific papers in this area

Author Response

ABSTRACT

Please change the order of the first two lines so it can have a better impact

  • We thank for the reviewer for this suggestion, and we have changed the order of the two lines in the revised manuscript.

What do the authors mean by “fabricating” in line 21…..seems disconnected from the last idea

  • We apologize for the lack of clarity due to the improper use of the word “fabricating.” We have removed the phrase “and cultivation/growing medium (“baglog”) fabricating“ to this end.

INTRODUCTION

Line 43-46 is good but is not in the proper place. Please re-order the referred idea to the final section of the introduction.

  • We agree with the reviewer’s suggestion and have moved this sentence to the final section of the Introduction.

Line 47, please add the surname of the author of reference 4 according to MDPI guidance

  • We have added the surname of the author in reference 4, as per the suggestion, in the revised manuscript.

Line 51, “however, their results”

  • We have replaced the word “study” with “their” in the revised manuscript.

METHODS

The methods are clear and easy to follow. However, a double-check is always preferred

  • We thank the reviewer for the suggestion. We have double-checked the Methods.

RESULTS

The addition of a picture of the experiment may improve how the readers can understand the mushroom production

  • We appreciate the reviewer’s suggestion, and we have added Figures 1 and 2 on page 5 to improve understandability.

DISCUSSION

The authors must stress the lack of information in this specific area, maybe with a short table or by adding a short paragraph that summarizes the number of scientific papers in this area

We thank the reviewer for the suggestion, and we have added a short paragraph at the end of the Discussion.

Reviewer 2 Report

Comments and Suggestions for Authors

The article “Aqueous Extracts of Lemon Basil Straw as Chemical Stimulator for Gray Oyster Mushroom Cultivation” is dedicated to obtained extracts as substrate for mushroom cultivation. I recommend revision with some improvements.

My recommendations:

  1. Please add some references to the phrase at line 57 “For the study, various physical and chemical treatments were demonstrated to promote mycelial growth and an increased production yield of edible mushrooms”.
  2. The Introduction section must be improved with more detailed about studied subject. Please add more than 5 new references in Introduction section. A correlation between the previous published paper, reference [4] and the present paper, maybe a continuous work of previous experiments, etc.
  3. The novelty of the paper should be underlined.
  4. Please add town, city to the equipment’s used, ex. GC-MS.
  5. For analysis of bioactive compounds, you use GC-MS for analysis. Please give more information about the following phrase “The aqueous extract was diluted in absolute ethanol (99.995%) to a concentration of 500 ppm before analysis was conducted, in order to reduce the interfering effect of moisture on the filament of the MS ion source.”. You make extraction with ethanol; a concentration of 500 ppm is very high for GC-MS analysis. Please give more details about the analytic method used: linearity range, calibration curve, etc. Also, at line 161 “The compounds were identified by comparison of sample mass spectrum using 161 the NIST2011 library” I believe you make a calibration curve or was only qualitatively analysis, because at lines 224-225 was affirmed “The concentration of 9-octadecenamide in the M-LBS extract (17.34 wt%) is lower than the D-LBS extract (59.99 wt%)”.
  6. Line 221 please correct Figure 1 with Figure 2 for 9-octadecenamide identification.
  7. Some journal names are missing from References section, ex. 14, 15. 16, 19, 20, 21, 22.
  8. I think that the paper can be improved, more detailed are needed in Results and Discussion section, only some results are given.

Author Response

  1. Please add some references to the phrase at line 57 “For the study, various physical and chemical treatments were demonstrated to promote mycelial growth and an increased production yield of edible mushrooms”.
  • We have added references 6–10 as recommended by the reviewer.
  1. The Introduction section must be improved with more detailed about studied subject. Please add more than 5 new references in Introduction section. A correlation between the previous published paper, reference [4] and the present paper, maybe a continuous work of previous experiments, etc.
  • To address this comment, we have added 5 new references [3–7], which are related to the subject of alternative substrates for mushroom cultivation.
  1. The novelty of the paper should be underlined.
  • We have highlighted the novelty of our work at the end of the Introduction.
  1. Please add town, city to the equipment’s used, ex. GC-MS.
  • Town and city information for the manufacturers of all analytical equipment have been added in the revised manuscript.
  1. For analysis of bioactive compounds, you use GC-MS for analysis. Please give more information about the following phrase “The aqueous extract was diluted in absolute ethanol (99.995%) to a concentration of 500 ppm before analysis was conducted, in order to reduce the interfering effect of moisture on the filament of the MS ion source.”. You make extraction with ethanol; a concentration of 500 ppm is very high for GC-MS analysis. Please give more details about the analytic method used: linearity range, calibration curve, etc. Also, at line 161 “The compounds were identified by comparison of sample mass spectrum using 161 the NIST2011 library” I believe you make a calibration curve or was only qualitatively analysis, because at lines 224-225 was affirmed “The concentration of 9-octadecenamide in the M-LBS extract (17.34 wt%) is lower than the D-LBS extract (59.99 wt%)”.
  • We apologize for the lack of clarity due to incorrect language. According to our chemical analyst, 500 ppm is not very high for GC-MS analysis, because it is 0.5%wt of the aqueous extract diluted in ethanol. We added “0.5 wt%” to line 165 of the revised manuscript. We did not make a calibration curve, and have not reported the linearity range. Instead, compound amounts were quantified by using the total sum normalization method [14] as mentioned in lines 167 and 168. In particular, the sentences “After 53 min of analysis, the column temperature and flow rate were ramped up to 300°C and 2 mL/min, respectively, for post treatment” and “…whereas the ion source voltage was set to 70 eV. Solvent delay was set at 2 min.” were added to lines 161 and 162 and 164 and 165, respectively.
  1. Line 221 please correct Figure 1 with Figure 2 for 9-octadecenamide identification.
  • We thank the reviewer for pointing out the error. Lines 221 and 242 have been corrected in the revised manuscript.
  1. Some journal names are missing from References section, ex. 14, 15. 16, 19, 20, 21, 22.
  • We have updated our references and added the journal names to all references.
  1. I think that the paper can be improved, more detailed are needed in Results and Discussion section, only some results are given.
  • We thank the reviewer for this suggestion. We have provided additional information at the end of the Discussion.

Round 2

Reviewer 2 Report

All the requirements were made. 

Author Response

The authors are thankful for helpful comments, which improved the manuscript.